# Lysyl-Oxidase Dependent Extracellular Matrix Stiffness in Hodgkin Lymphomas: Mechanical and Topographical Evidence

**DOI:** 10.3390/cancers14010259

**Published:** 2022-01-05

**Authors:** Massimo Alfano, Irene Locatelli, Cristina D’Arrigo, Marco Mora, Giovanni Vozzi, Aurora De Acutis, Roberta Pece, Sara Tavella, Delfina Costa, Alessandro Poggi, Maria Raffaella Zocchi

**Affiliations:** 1Division of Experimental Oncology and Unit of Urology, Urologic Research Institute, IRCCS Ospedale San Raffaele, 20132 Milan, Italy; alfano.massimo@hsr.it (M.A.); locatelli.irene@hsr.it (I.L.); 2Department of Electron Microscopy, Institute for Chemical Science and Technologies-National Research Council (SCITEC-CNR), 16149 Genoa, Italy; cristina.darrigo@ge.ismac.cnr.it; 3Pathology Unit, IRCCS Ospedale Policlinico San Martino, 16132 Genoa, Italy; marco.mora@hsanmartino.it; 4Department of Information Engineering, University of Pisa, 56122 Pisa, Italy; g.vozzi@ing.unipi.it; 5Research Center E. Piaggio, University of Pisa, 56122 Pisa, Italy; aurora.deacutis@for.unipi.it; 6Cellular Oncology Unit, Department of Experimental Medicine, IRCCS Ospedale Policlinico San Martino, University of Genoa, 16132 Genoa, Italy; roberta.pece@hotmail.com (R.P.); sara.tavella@unige.it (S.T.); 7Molecular Oncology and Angiogenesis Unit, IRCCS Ospedale Policlinico San Martino, 16132 Genoa, Italy; delfina.costa@hsanmartino.it (D.C.); alessandro.poggi@hsanmartino.it (A.P.); 8Division of Immunology, Transplants and Infectious Diseases, IRCCS San Raffaele Scientific Institute, 20132 Milan, Italy

**Keywords:** Young’s modulus, 3 dimensional cultures, non-Hodgkin lymphomas, mesenchymal stromal cells, Reed–Sternberg cells

## Abstract

**Simple Summary:**

Alterations of the composition and architecture of the extracellular matrix (ECM), leading to increased stiffness, is known to condition development, invasiveness and severity of neoplasms. In this study, we report increased lymph node (LN) stiffness in human lymphomas, measured by LN elastometry or by computerized imaging of bioptic specimens. Stiffness matched to lymphoma histotype and grading. The enzyme lysyl oxidase (LOX) is involved in the rise of collagen cross-linking in Hodgkin lymphomas, while altered architecture, shown by scanning electron microscopy and polarized light microscopy is involved in advanced follicular lymphomas. Based on these data, digital pathology may help in the staging of lymphomas, and lysyl oxidase may represent a target for therapy in Hodgkin lymphomas.

**Abstract:**

Purpose: The biochemical composition and architecture of the extracellular matrix (ECM) is known to condition development and invasiveness of neoplasms. To clarify this point, we analyzed ECM stiffness, collagen cross-linking and anisotropy in lymph nodes (LN) of Hodgkin lymphomas (HL), follicular lymphomas (FL) and diffuse large B-cell lymphomas (DLBCL), compared with non-neoplastic LN (LDN). Methods and Results: We found increased elastic (Young’s) modulus in HL and advanced FL (grade 3A) over LDN, FL grade 1–2 and DLBCL. Digital imaging evidenced larger stromal areas in HL, where increased collagen cross-linking was found; in turn, architectural modifications were documented in FL3A by scanning electron microscopy and enhanced anisotropy by polarized light microscopy. Interestingly, HL expressed high levels of lysyl oxidase (LOX), an enzyme responsible for collagen cross-linking. Using gelatin scaffolds fabricated with a low elastic modulus, comparable to that of non-neoplastic tissues, we demonstrated that HL LN-derived mesenchymal stromal cells and HL cells increased the Young’s modulus of the extracellular microenvironment through the expression of LOX. Indeed, LOX inhibition by β-aminopropionitrile prevented the gelatin stiffness increase. Conclusions: These data indicate that different mechanical, topographical and/or architectural modifications of ECM are detectable in human lymphomas and are related to their histotype and grading.

## 1. Introduction

The local microenvironment has an important role in the control of tumor development [1,2]; different cell types contribute to cancer progression, showing either permissive or suppressive effects [3,4,5]. In particular, mesenchymal stromal cells (MSC) can influence and shape the immune response against lymphomas, becoming a proposed target of therapy [6,7]. In turn, extracellular matrix (ECM) components also act as modulators of neoplastic progression, tissue invasion and metastasis [8,9,10]. Alterations of the biochemical composition or architecture of ECM is reported to condition the behavior of different solid neoplasms [10,11,12]. Some modifications, such as increased collagen production and cross-linking that initiate fibrosis, at first can limit tumor local invasion. However, maintenance of uncontrolled fibrotic processes can induce modifications of the three-dimensional (3D) structure of the tissue, whose remodeling eventually leads to spatial environmental changes that improve cancer-cell metabolism and favor tumor progression [8,9,10,11,12,13,14]. As an example, the metastatic potential of different carcinomas has been related to collagen IV or XII overexpression and dysregulation of ECM structure, causing increased tissue stiffness, altered tissue structure and tumor-cell dissemination [15,16]. Recently, a direct correlation was proposed between biochemical–ultrastructural ECM modifications and cancer invasiveness, contributing to the severity of disease [14,17,18].

From this viewpoint, the definition of the structural or architectural alterations of tumor stroma and ECM, including mechano-elastic features [19,20], may carry information on tumor aggressiveness and represent an additional tool to frame and target the stage of disease. The relevance of the microenvironment in lymphoma onset and maintenance is documented by histological and gene-expression studies [21,22,23,24], although the role of ECM in these processes is still to be defined.

In this study we investigated the degree of ECM stiffness, collagen cross-linking and anisotropy in the lymph nodes (LN) of lymphoma patients. Comparison of these parameters in Hodgkin lymphomas (HL), follicular lymphomas (FL), diffuse large B cell lymphomas (DLBCL) and lymphadenitis (LDN) revealed increased stiffness and elastic (Young’s) modulus in HL and advanced FL (grade 3A) over LDN and FL grade 1–2, with intermediate values in DLBCL. Larger stromal areas could be measured in HL by digital pathology and scanning electron microscopy (SEM); in turn, architectural modifications were documented by SEM in FL3A. Of note, increased collagen cross-linking was found in HL, while enhanced anisotropy was observed in FL3A. Interestingly, HL expressed high levels of lysyl oxidase (LOX), one of the main enzymes responsible for collagen cross-linking and matrix stiffness [25,26]. Next, by using gelatin scaffolds fabricated with a low elastic modulus, comparable to that of non-neoplastic tissues [26,27], we demonstrated that LN-derived mesenchymal stromal cells (LN-MSC) and HL cells increased the Young’s modulus of the extracellular microenvironment through the expression of LOX. Indeed, LOX inhibition by β-aminopropionitrile (BAPN) prevented the stiffness increase in repopulated gelatin scaffolds. These data indicate that different mechanical, topographical and/or architectural modifications of ECM are detectable in human lymphomas and are related to their histotype and grading.

## 2. Materials and Methods

### 2.1. Patients

LN biopsies and sections of patients with Hodgkin lymphoma (HL, *n* = 11) or non-Hodgkin lymphomas (NHL, *n* = 24) were obtained from the Pathology Unit, IRCCS Ospedale Policlinico San Martino, Genoa, under conventional diagnostic procedures, with informed consent and approval by the institutional ethical committee (IRB approval 0026910/07, 03/2009, and 14 September 2015). Anonymized patient data were analyzed under a virtual private network (VPN) with password-database access limited to authorized operators belonging to the Molecular Oncology Unit of the Ospedale Policlinico San Martino. NHL were subdivided by histotype and grading into follicular lymphomas grade 1–2 (FL1-2, *n* = 6), grade 3A (FL3A, *n* = 10) and diffused large B cell lymphomas (DLBCL, *n* = 8) (Table 1) [28].

Ten reactive LN were also analyzed (LDN). ECM from the LN was prepared according to a decellularization protocol previously described [11]. Slides of hematoxylin-eosin (HE) or Mallory trichrome staining from LN sections of each patient were subjected to the Digital Patology Imaging and immunohistochemistry performed as described in the relative section.

### 2.2. Immunohistochemisty (IHC)

3D cultures were fixed in HistoChoice (Amresco, Solon, OH, USA) overnight at 4 °C. After dehydration in ethanol, samples were clarified in xylene and paraffin embedded. For each scaffold 5 µm serial sections were cut up to a depth of 50 µm, dried overnight at 37 °C and analyzed by IHC. Sections (5 µm) of LN specimens were obtained from the Pathology Unit of the IRCCS Ospedale Policlinico San Martino. IHC was carried out using the automated stainer BOND RXm (Leica Biosystem, Nussloch GmbH, Germany), according to the manufacturer’s instructions, after sections deparaffinization at 72 °C for 30 min, as follows. For LOX/TG2 diaminobenzidine (DAB) detection: 1. antigen retrieval with the ER1 solution at 95 °C for 10 min; 2. Perox-Blok, 10 min at room temperature (RT); 3. rabbit anti-human LOX mAb (1:900 dilution, ab174316 Abcam) or anti-TG2 mAb (2 µg/mL, TGM2, Thermo Fisher Scientific, Rochester, NY, USA), 20 min at 37 °C; 4. Post Primary reagent (Leica) 8 min; 5. Polymer (Leica) 8 min; 6. Mixed DAB Refine (Leica) 8 min; 7. counterstaining with Hematoxilin and mounting with Mount Quick aqueous (Bio Optica, Milan, Italy). For automated multiplex IHC, the rabbit anti-human CD30 (1:10 dilution, BerH2 clone, Biocare Medical, Pacheco, CA, USA), the rabbit anti-MUM-1 (undiluted, MRQ-43 Cell Marque Corporation, Rocklin, CA, USA) and the rabbit anti-human LOX mAb were used. The Bond Polymer Refine Detection kit was used to detect CD30 or MUM-1 following the protocol steps from 1 to 6 as described above. LOX was evidenced using the Vina Green Chromogen (Biocare Medical, Rocklin, CA, USA) applied in the Leica Research Kit according to the manufacturer’s instructions. Briefly: 1. incubation (10 min) with Wash Buffer; 2. rabbit anti-human LOX mAb (1:900 dilution) for 20 min at RT; 3. incubation (8 min) with Post Primary, followed by 8 min with Polymer; 4. mixed Vina Green Chromogen detection for 10 min at RT; 5. counterstaining with Hematoxilin and mounting with Mount Quick aqueous. For LOX alkaline phosphatase (AP) detection: 1. antigen retrieval with the ER1 solution at 95 °C for 10 min; 2. rabbit anti-human LOX mAb (1:900 dilution) 20 min at 37 °C; 3. Post Primary AP reagent 9 min; 4. Polymer-AP 15 min; 5. Mixed RED Refine 8 min; 6. counterstaining with Hematoxilin and mounting as above.

### 2.3. Digital Pathology Imaging

Digital images were acquired using the Aperio ScanScope Slide program of the Aperio AT2 Scanner (Leica Biosystem, Aperio Technologies, Nussloch GmbH, Germany) at 20× magnification. The pattern recognition tool of the Genie Classifier software (Leica Biosystem), excluding glass or fat regions, was used to determine the percentage area of each compartment per slide. Software training was performed using a montage of example images manually marked by the operator to distinguish the individual areas of interest (Appendix A). Based on manufacturer recommendations, a 95% or greater training accuracy was required for each classifier in order to proceed with image analysis of each sample. Tissue-area percentage was adjusted to total tissue area (mm^2^) in the image analyzed, ignoring the glass. All images were controlled for accuracy of the tissue recognition software, in addition to accuracy of the image analysis algorithm. Two pathologists checked that stromal areas were identified correctly. Mallory trichrome images were captured with the AperioAT2 scanner at 20× magnification, viewed and annotated with the Aperio Image-Scope program (Leica Biosystem, Aperio Technologies). Layers of each virtual slide annotation were created manually, selecting the whole area, and analyzed with the automated image analysis algorithm color deconvolution (Aperio software version 9.1, Aperio Technologies, Nussloch GmbH, Wetzlar, Germany). The thresholds were set for the identification of blue for fiber component, violet for nuclei and yellow for red blood cells [29].

### 2.4. Scanning Electron Microscopy (SEM)

ECM were fixed with 4% paraformaldehyde in phosphate buffer (PB) for 5 h, postfixed in 1% osmium tetroxide in PB for 1 h, dehydrated in ascending degree of ethanol (70–80–95–100%), dried for 30 min in xylene and embedded with paraffin after evaporation of xylene and serially cut as 10 μm thick sections using a Leica microtome. Sections were first deparaffinized in xylene for 2 h and then hydrated in descending degree of ethanol (100–95–80–70%) and H_2_O. Ultrathin sections were collected on glass coverslips, mounted on aluminum stubs and sputter-coated 3 min with gold. Morphology and ultrastructure were investigated by SEM on a TM3000 Benchtop SEM (Hitachi, Pleasanton, CA, USA) instrument operating at 15 kV acceleration voltage [30]. Areas of matrix or empty spaces were calculated with a tool of the ImageJ software (version 2.0, National Institute of Health, Bethesda, MD, USA) and reported as percentage area on the whole image analyzed. Results are also reported as perimeter/area ratio of empty spaces. Repopulated GelMA scaffolds were analyzed by SEM as above, after fixation without embedding.

### 2.5. Collagen Cross-Linking and Anisotropy Evaluation

ECM samples were hydrolyzed in 1 mL of 6 M HCl for 20 h at 110 °C, and the acid lysate added of internal standard control and assayed for hydroxyproline content by HPLC (Hydroxyproline reagent kit 195-9501, Bio-Rad, Milan, Italy); the amount of collagen was estimated based on the level of hydroxyproline accounting for 13.5% of the collagen amino acid composition. Hydroxylysylpyridinoline (HP) and lysylpyridinoline (LP) were measured by HPLC, in ECM hydrolized in 0.5 mL of 6 M HCl for 20 h at 110 °C, and added of internal standard (Crosslinks, pyridinolin and deoxypyridinolin kit 48000, Chromsystem Biochemicals GmbH, Gräfelfing/Munich, Germany) [11]. The degree of organization of fibrils (anisotropy) was evaluated using 3 areas from the same histological section, using 10× magnification. Imaging from the histological section was acquired on brightfield to show the tissue location, and on darkfield and polarized light to obtain collagen birefringence. Anisotropy was estimated by using the function “measure” and the plug-in FibrilTool in the ImageJ software. Anisotropy was measured on the entire area of the section, in order to avoid any bias due to the selection of areas excluding or over-representing certain fibers in the outputs [31].

### 2.6. Scaffold Fabrication and Young’s Modulus Measurement

Gelatin methacrylate (GelMA) scaffolds were fabricated and mechanically characterized according to the following procedures. GelMA was synthesized as described [32]. Briefly, gelatin (type A, porcine skin, Sigma-Aldrich, St. Louis, MO, USA) was stirred at 10% (*w*/*v*) into Dulbecco’s phosphate buffered saline (DPBS; P0750, Biowest, Nuaillé, France) at 50 °C for 1 h. Subsequently, methacrylic anhydride 10% (*v*/*v*) (Sigma-Aldrich) was added dropwise at a rate of 0.5 mL/min to the gelatin solution while stirring at 50 °C, and it was allowed to react for 1 h at 50 °C: The obtained GelMA solution was therefore 5× diluted and dialyzed against deionized water by using 12–14 kDa cutoff dialysis tubing (Sigma-Aldrich) for one week at 40 °C. The solution was finally freeze-dried (BenchTop Pro-SP Scientific, Stone Ridge, NY, USA) at −60 °C for 24 h, and stored at −20 °C until further use. GelMA (10% *w*/*v*) was fully dissolved in DPBS at 50 °C containing 0.5% (*w*/*v*) 2-hydroxy-4′-(2-hydroxy ethoxy)-2-methylpropiopheone (Irgacure 2959, Sigma-Aldrich), as photoinitiator. The prepolymer was poured into a Petri dish (20 mL of prepolymer into 80 mm diameter Petri dish) and cooled down at 4 °C for 10 min. Finally, it was photo-crosslinked for 30 min in a UV crosslinking box (UVP CL-1000L Ultraviolet Crosslinker) at 365 nm, 12 mW/cm^2^ at room temperature. The photo-crosslinked GelMA was frozen at −20 °C and finally freeze-dried (BenchTop Pro-SP Scientific, Stone Ridge, NY, USA) at −60 °C for 12 h. Microporous cylindrical samples (Ø = 5 mm, thickness = 3 mm) were cut and stored at room conditions until further use. Uniaxial compressive load–unload cycles tests were performed on GelMA scaffolds using a Z005 series Zwick/Roell testing machine endowed with a 100 N load cell. Wet samples were soaked in ultrapure water at 37 °C until reaching the weight equilibrium. Samples were tested at a strain of 0.01 s^−1^ to a compressive axial deformation of 30% [27]. The Young’s modulus of each sample was evaluated from the slope of the initial linear portion of the stress–strain curve. At least 10 specimens for condition were tested. Some samples of fresh LN bioptic specimens or repopulated scaffolds underwent the same analysis to determine the elastic properties as Young’s modulus.

### 2.7. Three-Dimensional (3D) Cultures of Repopulated GelMA Scaffolds

GelMA scaffolds were dipped in MSC medium, i.e., 1 mL MEM-α containing desossiribonucleotides supplemented with 2 mM glutamine, 1% penicillin/streptomycin (GIBCO, Thermo Scientific, Rochester, NY, USA) and 1% Chang medium (Irvine Scientific, Wicklow, Ireland), 1 h before cell seeding [30,33]. A mixture of 3 × 10^5^ L428 HL cells (DSMZ GmbH, Braunschweig, Germany) in 100 µL of RPMI1640 medium and 2 × 10^5^ HL-derived LN-MSC, obtained as described [33], in 100 µL of MSC medium were seeded onto the pre-wet GelMA scaffolds in a 24 w plate and kept 2 h at 37 °C in a 5% CO_2_ humidified incubator, then 1 mL of MSC/RPMI1640 (50% *v*/*v*) medium was added and samples cultured up to 14 d. To avoid cell starvation, half medium was replaced every two days. At 7 d and 14 d, repopulated GelMA scaffolds were fixed in 4% buffered formalin or Histochoice and embedded in paraffine for IHC. Parallel samples underwent Young’s modulus measurement and SEM analysis.

### 2.8. LOX Inhibition and Cell Viability Assays

The LOX inhibitor BAPN (A-3134 Sigma) was used at serial dilutions (100 mM to 50 µm) to determine the maximum nontoxic concentration on LN-MSC and L428 HL cells. LN-MSC viability was determined with Crystal Violet Assay Kit (Biovision, Milpitas, CA, USA): the amount of crystal violet proportional to the number of living cells was measured with the VICTORX5 plate reader (Perkin Elmer, Milan, Italy) at the optical density (O.D.) of 595 nm [34]. Since the crystal violet assay can be performed on adherent cells only, L428 cell viability was determined with the CellTiter-Glo^®^ Luminescent Cell Viability Kit (Promega Italia Srl, Milan, Italy) using the luciferase reaction consisting in mono-oxygenation of luciferin catalyzed by luciferase in the presence of Mg^2+^, ATP and molecular oxygen [33]. Luminescence was detected with the VICTORX5 plate reader and expressed as luminescence arbitrary units (a.u.). Three logarithmic BAPN dilutions were chosen for functional experiments on repopulated scaffolds. LOX enzymatic activity was assessed in supernatants (SN) recovered from 50 × 10^4^ LN-MSC or L428 cells cultured for 48 h in RPMI1640, without phenol red, with B27 supplement (Thermo Fisher), with the specific fluorometric assay (Ex/Em = 535/587 nm, BioVision, Milpitas, CA, USA) [35]. LOX activity was evaluated as delta (Δ) mean fluorescence intensity (arbitrary units, a.u.) measured at t60–t0. Glucose was measured in the SN recovered from repopulated scaffolds every 48 h up to 10 d, untreated or treated with BAPN as above, with the D-glucose Assay Kit (Megazyme Int. Ltd., Wicklow, Ireland) and referred to a standard curve. Data expressed as percentage glucose consumption vs. glucose content in culture medium.

### 2.9. Statistical Analysis

Data are presented as mean ± SEM (standard error of the mean). Statistical analysis was performed using ANOVA, or the two-tailed unpaired Student’s *t* test, with Welch correction, using the GraphPad Prism software 5.0 (GraphPad Software Inc.; San Diego, CA, USA). Correlation analysis was performed applying the Pearson’s r coefficient to data showing normal distribution.

## 3. Results

### 3.1. Detection of Larger LN Stromal Areas in HL vs. NHL by Digital Pathology Imaging

Slides of HE or Mallory trichrome staining from LN sections of patients with LDN, HL, FL1-2, FL3A, DLBCL (Table 1) were subjected to the Digital Pathology Imaging. HE stained sections were analyzed with the Genie software (Genie Classifier, Leica Biosystem) to determine the percentage area of stromal compartment per slide, adjusted to total tissue area in the image analyzed (Appendix A). As shown in Figure 1A,C, the percentage of stromal areas was significantly higher in HL (mean 15%) than in FL1-2 or LDN (mean 8%), but not in DLBCL and FL3A (mean 12% and 6%, respectively). To better distinguish stromal areas, digital images of Mallory trichrome (collagen specific) stained sections were analyzed with an automated image analysis algorithm for color deconvolution that provides information on area and intensity of staining for each individual stain, avoiding cross-contamination [29]. ImageScope software was then applied to quantify Mallory trichrome staining (Appendix A). Figure 1B,D confirm, with more precision, the presence of significantly larger collagen areas in HL samples (mean 16%) compared to FL1-2 (mean 8%) or LDN (mean 7%); by this staining specific for collagen, DLBCL and FL3A revealed a higher, although nonsignificant percentage of stromal areas (10% and 9%, respectively).

Thus, digital pathology can measure the extension of stromal areas in LN from lymphoma patients and underline differences related to the disease, compared to LDN, histological type (HL vs. DLBCL) or grading inside a definite histological type (FL3A vs. FL1-2).

### 3.2. Different Ultrastructure of Lymph Node ECM in HL and NHL

To gain insight into morphology and ultrastructure, ECM obtained from the LN of HL and NHL were analyzed by SEM. The morphology of ECM was heterogeneous, apparently related to the histologic type: in particular, interconnected stromal branches, leaving round or oval well-separated empty spaces, remaining after the removal of cells, were detectable in LDN (Figure 2A). Conversely, in FL3A, and to a lesser extent in FL1-2 and DLBCL, ECM branches were shorter, with breaks and spikes, and empty spaces were reduced and irregular. In FL3A, fibers appeared thick and twisted; in turn, ECM from HL appeared dense and compact, with very few gaps and matrix organized in tight bundles (Figure 2A). Empty areas were then calculated with the ImageJ software and reported as percentage area on the whole image or as perimeter/area (P/A) ratio of empty spaces. In HL, the percentage of empty areas was significantly reduced (with reciprocal increase of matrix areas) (Figure 2B); in addition, a significant increase in the P/A ratio of gaps, indicating their ellipsoidal, rather than oval or round shape, was evidenced in FL3A samples and, to a lesser extent, in DLBCL (Figure 2C).

These data point to an altered structure and architecture of ECM in lymphomas, compared to non-neoplastic LN (LDN), indicating an increase in the compactness of ECM in HL and shape alteration in FL3 and DLBCL.

### 3.3. Increased Collagen Cross-Linking in HL and Anisotropy in NHL

Due to the larger extension and altered architecture of ECM in HL and FL3A and DLBCL, compared to other lymphomas or LDN, increased stiffness of LN in these histological types can be forecasted. To verify this hypothesis, LN from HL, FL1-2, FL3A, DLBCL and LDN underwent elastometric measure. Figure 3A shows that the Young’s modulus of FL3A (mean 48 kPa), HL (mean 38 kPa) and, to a lesser extent, of DLBCL (mean 25 kPa) was higher than that of FL1-2 and LDN (mean 10 and 4.7 kPa, respectively), indicating increased LN stiffness in lymphomas, related to histological type and grading. To investigate the nature of stiffness, collagen structure was examined by measuring the cross-linking of hydroxylysyl-pyridinoline (HP) and lysyl-pyridinoline (LP); these modifications are due to oxidative deamination of hydroxylysine and lysine by lysyl oxidase (LOX) in the presence of oxygen [25,26]. A significant enhancement in the amount of collagen cross-linking was found in the ECM of HL vs. DLBCL, FL1-2 and LDN, while FL3A showed heterogeneous amounts (Figure 3(Ba)); the HP/LP ratio was also increased in HL, at variance with other lymphomas (Figure 3(Bb)). In turn, as an alternative explanation for increased stiffness, the linearization of collagen fibers evidenced by polarized light microscopy (Figure 3C) was significantly higher in FL3A than in other lymphomas or LDN (Figure 3D).

These data indicate that several modifications of the ECM apply to lymphomas. Physical and ultrastructural modifications were evident in HL and FL3A LN; in HL the increased stiffness was associated with a modified structure of the ECM (crosslinking), whereas in FL3A the increased stiffness was associated with an increased linearization of the collagen fibers (anisotropy).

### 3.4. High LOX Expression in HL Specimens

Since LOX is reported as the major enzyme responsible for collagen structure remodeling toward an increased stiffness [25,26], LN sections from HL vs. NHL patients were examined by IHC for LOX expression and compared to collagen staining. Figure 4 shows that LOX is clearly detectable in HL sections, unlike FL3A (Panel A: conventional Mallory staining and digital imaging; Panel B: deconvoluted images showing collagen in blue and LOX in brown), and FL1-2 or DLBCL (Appendix A). The analysis of 10 HL, 6 FL3A, 6 FL1-2 and 6 DLBCL and 6 LDN, evidenced that LOX expression in HL is significantly higher than in FL1-2 (*p* < 0.005) or DLBCL and LDN (*p* < 0.05); this difference was detectable, although not statistically significant, also vs. FL3A (Figure 4C).

Of note, HL cells identified as CD30^+^ (Figure 4D,E, subpanels a, two representative cases), expressed intracellular LOX that was not detectable in non-neoplastic areas (subpanels d). MUM-1/LOX double staining, performed to detect nuclear MUM-1 and cytoplasmic LOX avoiding overlapping, confirmed LOX localization in HL (Figure 4D,E, subpanels b). In addition, Reed–Sternberg cells, which can be distinguished by morphology (Figure 4E, subpanels b and c, enlarged in e and f) display a strong LOX staining. In some areas extracellular LOX can also be distinguishable. Figure 5 shows a third HL (23014-18), where LOX was clearly distributed in stromal areas (Panel Ac), also recognized by Mallory staining for collagen (Panel Aa) and TG2 staining for stromal cells (Panel Ab). Deconvoluted images of Mallory, TG2 and LOX staining are shown in Figure 5B (subpanels a, b and c, respectively).

In addition, in this HL case, CD30^+^ (Panel C, subpanels a and b) and MUM-1^+^ cells (Panel Cc) contain variable amount of LOX that can also be detectable as extracellular in some areas.

Figure 6 depicts the significant correlation between Young’s modulus and collagen crosslinking (Panel Aa) or LOX expression (Panel Ab) in HL, at variance with NHL FL3 (Panels Ba and Bb) where the correlation is significant only between Young’s modulus and anisotropy index (Panel Bc), that is not the case of HL (Panel Ac). This would support the hypothesis that LOX directly contributes to HL collagen stiffness.

### 3.5. Involvement of LOX in ECM Stiffness

To demonstrate a direct role of LOX in the increase of ECM stiffness in HL, a 3D culture model was set up, using GelMA scaffolds fabricated to display a Young’s modulus similar to non-neoplastic LN. GelMA was chosen due to its intrinsic hydration and mechanical characteristics, tunable through methacrylation degree and gelatin concentration, that allow cultured cells to modify the structure of this peculiar biomaterial [32]. cell-responsive microtissues GelMA scaffolds were repopulated with LN-MSC or with a mixture of LN-MSC and L428 HL cells, since both cell types can produce the enzyme. After 7 and 14 days of culture, scaffolds were efficiently repopulated as shown by SEM (Figure 7A, subpanels a–c: 7 d, subpanel d: 14 d); IHC of sections of these 3D culture models demonstrated that both cell types colonized the scaffolds and LOX was detectable inside the 3D cultures (Figure 7B,C, brown staining), also in areas enriched with CD30^+^ L428 HL cells (Figure 7D a–c, LOX green staining). The Young’s modulus of empty GelMA scaffolds was comparable to that of non-neoplastic LN (<10 kPa, compare CTR in Figure 6E with LDN in Figure 3A). Interestingly, after culture with LN-MSC, the Young’s modulus of GelMA scaffolds increased in 2 weeks, with a significant peak at 7 d (Figure 7E); the addition of L428 cells did not enhance the effect of LN-MSC, suggesting that the latter cell type is mainly responsible for matrix hardening, although also HL cells are able to produce LOX. To verify whether this enzyme was indeed engaged in the process of collagen stiffening, the LOX inhibitor BAPN [36] was added to the 3D cultures at different concentrations (from 5 mM to 50 µm) and Young’s modulus measured at 7 d.

Titration experiments were performed, since BAPN effective concentrations reported in the literature varied from 10 mM to 10 µm, depending on cell type (solid tumors or cancer associated fibroblasts) and on culture conditions (conventional cultures vs. spheroids) [36,37,38]. The concentrations chosen for our experimental setting were not toxic for either LN-MSC or L428 cells (Appendix A), but were efficient in inhibiting LOX activity in both cell types (Appendix A). As shown in Figure 7F, a dose-dependent reduction of the increase of GelMA stiffness induced by LN-MSC, or the mixture of LN-MSC + L428 cells, was detected by adding BAPN at 5 or 0.5 mM concentration during the culture. These data support the hypothesis that increased stiffness consequent to enhanced collagen cross-linking in HL, are conceivably due to sustained activity of LOX, produced mainly by LN-MSC and in part by CD30^+^ HL cells.

## 4. Discussion

Herein we report that Young’s (elastic) modulus, an index of stiffness, is higher in HL than in NHL LN, due to enhanced collagen cross-linking; in turn, NHL FL3A show increased elastic modulus compared to FL1-2 or DLBCL, together with architectural modifications related to increased anisotropy. Of note, HL express high levels of LOX, one of the main enzymes responsible for collagen cross-linking and matrix stiffness [26]. Accordingly, LN-MSC and HL cells, both producing LOX, are able to rise the stiffness of GelMA scaffolds fabricated with low Young’s modulus, comparable to that of non-neoplastic tissues [27], and this effect is prevented by the LOX inhibitor BAPN.

This work provides evidence that different mechanical, topographical and architectural modifications of ECM are detectable in human lymphomas and can be matched to their histotype and grading. Although, it should be taken into account that tissues removed from the in vivo environment would undergo alterations in their physical and mechanical properties, the reported differences are referred to samples of lymphoma specimens always compared to non-neoplastic lymph nodes in the same conditions. In particular, while collagen crosslinking is apparently the main cause of increased ECM stiffness in HL, FL3A display architectural changes possibly contributing to ECM physical alterations. As these alterations can be evidenced and measured by computerized imaging of LN sections stained with conventional methods, digital pathology may offer a reliable tool to identify tissue characteristics related to disease progression. The importance of measuring tissue elastic properties in clinical practice is also documented by the increasing interest in ultrasound elastography, which may help in the staging of tumors [39]. Gradient of stiffness, amount of collagen cross-linking and ECM texture have all been described as important factors contributing to solid cancer-cell spreading and growth [8,14], but little has been documented so far in hematological neoplasia.

The factors contributing to the increased stiffness observed in DLBCL are still to be defined: indeed, neither collagen cross-linking nor anisotropy alteration are sufficient to unfold this finding. There are different possible explanations for this observation: first, stiffness in DLBCL is not directly related to LN stroma, but rather to the degree of lymphoma cell infiltration (overcrowding and indirect effect on the stroma due to cell traction [40]) within the LN, potentially leading to disruption of the follicular structures in some areas as reported in breast tumor [41].

Other collagen abnormalities, different from cross-linking and not detectable with polarized light, are responsible for shape alterations in DLBCL or other matrix proteins are damaged/modified. In any case, elastometry can evidence these alterations in all aggressive lymphomas.

The second point is the involvement of LOX in HL LN rigidity. In solid tumors, LOX is described as the major factor responsible for ECM remodeling: in the presence of oxygen the enzyme catalyzes oxidative deamination of lysine and hydroxylysine residues, leading to HP and LP crosslinks in collagen and enhancing tissue rigidity [11,36,38,42]. In HL, the increased HP/LP ratio matches with higher Young’s modulus; accordingly, LOX was mainly detectable in HL LN stromal areas, compared to NHL, where TG2^+^ MSC are detectable, and also Reed–Sternberg cells express cytoplasmic LOX. Of note, in a 3D model based on repopulated gelatin (GelMA) scaffolds fabricated to reproduce the Young’s modulus of non-neoplastic LN, HL-derived LN-MSC are able to enhance significantly the stiffness of these scaffolds. This effect can be prevented by the LOX inhibitor BAPN, suggesting that also in HL LOX can be proposed as a therapeutic target, such as in solid tumors [38,42]. The contribution of Reed–Sternberg-derived LOX seems to be marginal, possibly due to the low number of these cells in definite lymphoma areas, as their distribution is generally scattered, rather than concentrated in aggregates. Moreover, not all lymphoma cells in HL are Reed–Sternberg, as documented by the IHC showing double staining for CD30/LOX or MUM-1/LOX. Different cell types, including inflammatory cells, can condition tumor environment and influence ECM modifications [43]; this is particularly true in HL, where neoplastic cells often represent a minority of the tumor population [21,23,24,25]. How much is the relative contribution of the immune infiltrate, of the resident nontransformed cells and that of tumor cells is difficult to be assessed. Metalloproteases or inflammatory cytokines, such as tumor necrosis factor-α or transforming growth factor-β, frequently found in HL microenvironment, are reported regulators of ECM remodeling, directly or indirectly influencing LOX activity and collagen cross-linking [21,25,44].

## 5. Conclusions

In conclusion, different mechanical, topographical and/or architectural modifications of ECM are detectable in human lymphomas and matched to their histotype and grading.

Of note, these alterations are related to LOX activity in HL, and this enzyme may be considered a target for therapy at least in HL. Preclinical studies and a number of clinical trials based on the targeting of LOX by competitive inhibitors of blocking antibodies are already ongoing in solid tumors [15,38,44]. The reported evidence of gene expression signatures related to the ECM in lymphomas [22,23,24], together with our data on the overexpression of LOX in HL, can extend this therapeutic approach to hematologic neoplasia. Lysyl oxidase-like 2 inhibitors and the blocking antibody Simstuzimab are already in phase II clinical trials for fibrotic diseases, and some orally administered small inhibitors recently concluded Phase 1 and Phase 2a clinical trials in myelofibrosis patients intolerant or unresponsive to conventional approved treatments [45,46].

Finally, the GelMA 3D culture system can represent a useful tool to study both pathologic modifications of ECM and drug response in HL and NHL. In particular, GelMa intrinsic hydration and mechanical characteristics, tunable through methacrylation degree and gelatin concentration, may serve to develop good cell-responsive microtissues for in vitro pathogenetic studies and anticancer drug testing.

## Figures and Tables

**Figure 1 cancers-14-00259-f001:**
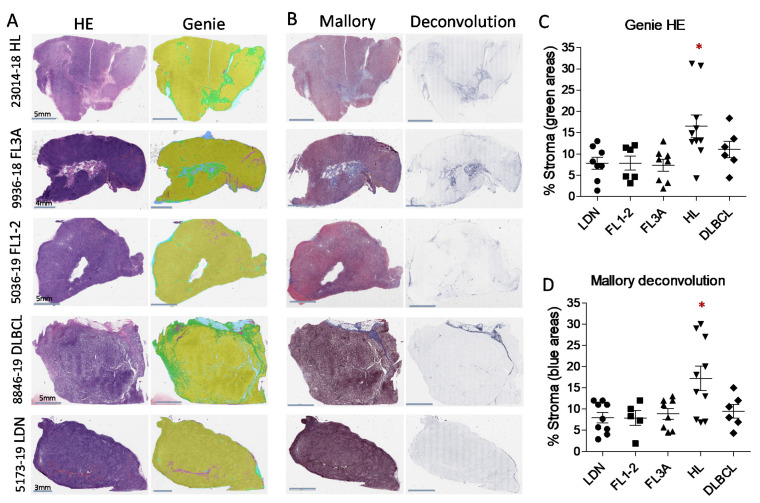
Digital pathology imaging and measurement of lymph node stroma in HL and NHL. Slides of hematoxilin-eosin (HE, representative cases in Panel (**A**)) or Mallory trichrome staining (representative cases in Panel (**B**)) from LN sections of patients with HL (*n* = 11), FL1-2 (*n* = 6), FL3A (*n* = 8), DLBCL (*n* = 6), compared with 9 LDN, were subjected to Digital Pathology Imaging. Panel (**C**). HE sections were analyzed with the Genie software (right images in Panel (**A**)) and the percentage area of stromal compartment per slide was calculated, adjusted to total tissue area in the image analyzed. Data are expressed as a percentage of green pseudocolor (stromal) areas and are the mean ± SEM of the indicated number of cases for each histological type. * *p* < 0.05 vs. LDN, FL1-2 and FL3A. Panel (**D**): digital images of Mallory trichrome stained sections were analyzed with an automated image analysis algorithm for color deconvolution (right images in Panel (**B**)) and collagen areas quantified by the Image Scope software. Data are expressed as percentage of blue (collagen) areas and are the mean ± SEM of the indicated number of cases for each histological type. * *p* < 0.005 vs. LDN, FL1-2 and FL3A.

**Figure 2 cancers-14-00259-f002:**
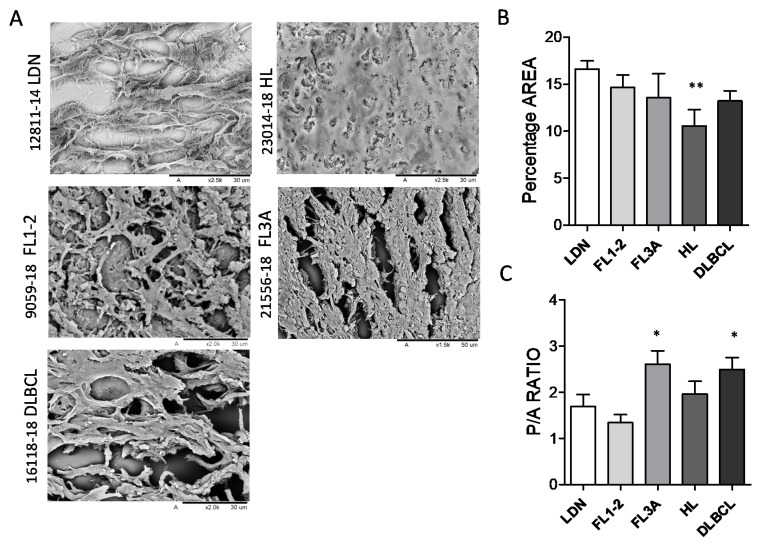
Ultrastructure of lymph node ECM in HL and NHL. ECM samples from HL (*n* = 6), FL1-2 (*n* = 6), FL3A (*n* = 6), DLBCL (*n* = 6) and LDN (*n* = 6) were paraffin embedded, serially cut as 10 μm thick sections and analyzed by SEM (Hitachi TM 3000). Panel (**A**): Sample slices of representative LDN, HL, FL1-2, FL3A and DLBCL Bar in each image: 30 μm. Panel (**B**,**C**): Areas of matrix or empty spaces were calculated with a tool of the ImageJ software and reported as percentage area on the whole image analyzed (**B**) or as perimeter/area (P/A) ratio of empty spaces (**C**). Mean ± SEM from 6 images analyzed for each histological type. Panel B: ** *p* < 0.005 vs. LDN, 0.05 vs. FL1-2 and FL3A. Panel (**C**): * *p* < 0.05 vs. LDN, FL1-2 and HL.

**Figure 3 cancers-14-00259-f003:**
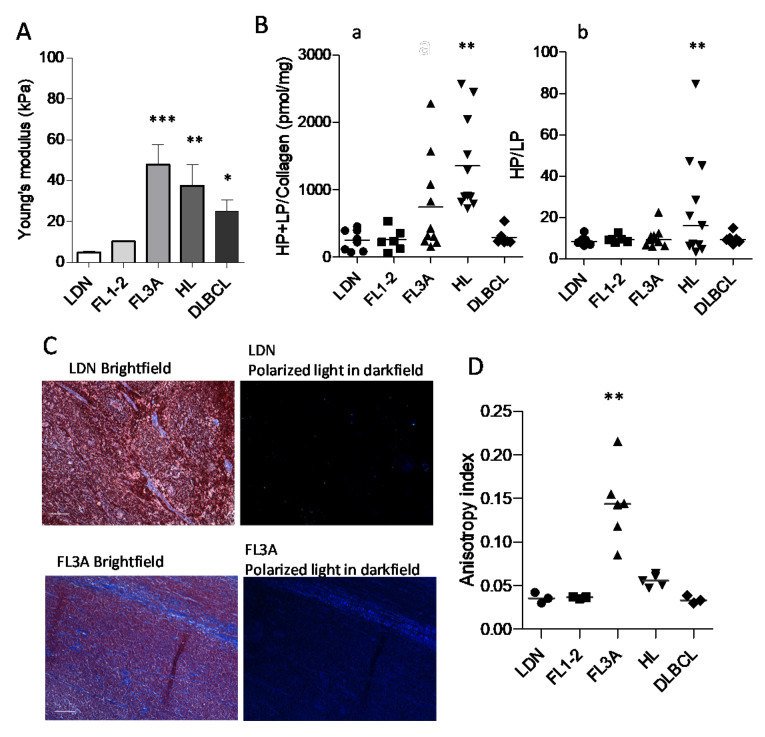
Young’s modulus, collagen cross-linking and anisotropy in HL vs. NHL. Panel (**A**): LN from HL (*n* = 6), FL1-2 (*n* = 4), FL3A (*n* = 4), DLBCL (*n* = 5) and LDN (*n* = 4) underwent elastometric measure (Young’s modulus) by compressive load-unload cycles (Zwick/Roell Z005). Results are expressed as kPa. Error bars are calculated as SD from the median. *** *p* < 0.0001 ** *p* < 0.001 and * *p* < 0.01 vs. LDN and FL1-2. Panel (**B**): ECM collagen structure of HL (*n* = 11), FL1-2 (*n* = 6), FL3A (*n* = 10), DLBCL (*n* = 6) and LDN (*n* = 8) as the cross-linking of HP and LP (**a**) and the ratio between HP/LP (**b**). Subpanel a: ** *p* < 0.001 vs. LDN, FL1-2 and DLBCL. Subpanel b: ** *p* < 0.001 vs. LDN, FL1-2, FL3A and DLBCL. Panel (**C**): Representative imaging from bright-field showing tissue location (Mallory staining for collagen) and dark-field showing collagen birefringence, for LN with LDN (16967-17) or FL3A (21556-18). Bar: 100 µm. Panel (**D**): Quantification of collagen fibers orientation, evaluated with the ImageJ software on 3 independent histological sections for each donor HL (*n* = 5), FL1-2 (*n* = 3), FL3A (*n* = 6), DLBCL (*n* = 3) and LDN (*n* = 3) at 10× magnification, with red bars indicating median values for each histotype. ** *p* < 0.001 vs. FL1-2 and DLBCL, *p* < 0.005 vs. HL and LDN.

**Figure 4 cancers-14-00259-f004:**
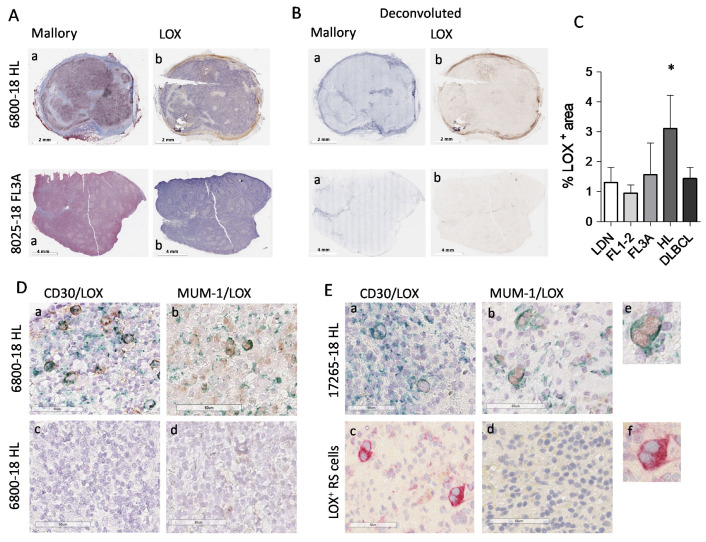
Digital pathology imaging of lymph node stroma and LOX expression in HL vs. NHL. Digital images of lymph node sections (6800-18, HL and 8025-18, NHL FL3) were acquired using the Aperio ScanScope Slide program of the AperioAT2 Scanner at 20× magnification. Panel (**A**): Subpanel (**a**): Mallory trichrome images. Subpanel (**b**): IHC with the anti-LOX rabbit ab174316 mAb Panel (**B**): Layers of each virtual Mallory slide (**a**) or LOX IHC stained slides (**b**) deconvoluted with the automated image analysis algorithm (Aperio version 9.1). Panel (**C**): LOX^+^ areas analyzed with the Aperio algorithm were quantified by the ImageScope software. Data are expressed as percentage of LOX^+^ areas and are the mean ± SEM of 10 HL and 6 cases for each of the other histological type. * *p* < 0.005 vs. FL1-2, *p* < 0.05 vs. LDN and DLBCL. Panel (**D**,**E**): digital images of sections from the HL lymph nodes 6800-18 and 17265-18, double-stained with the anti-CD30 BerH2 mAb (subpanels (**a**)) or the anti-MUM-1 mAb (subpanels (**b**)) followed by the specific GAM and DAB (brown) and the anti-LOX rabbit ab174316 mAb followed by GAR and Vina Green Chromogen (green). Subpanel Dc: negative control with the second reagents alone. Subpanel Ec: anti-LOX rabbit ab followed by GAR and AP (red). Subpanels d: MUM-1/LOX double staining in a non-neoplastic area. (**Ee**): enlargement of the square in (**Eb**,**Ef**): enlargement of the square in (**Ec**).

**Figure 5 cancers-14-00259-f005:**
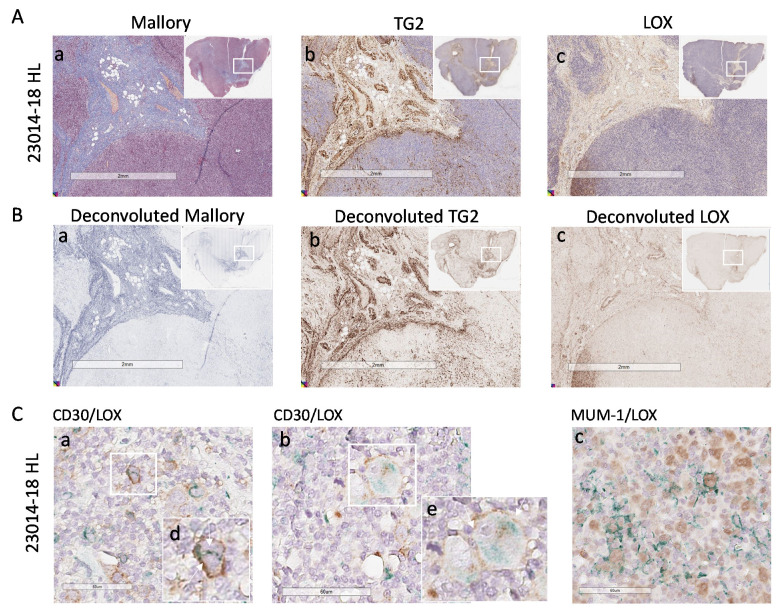
LOX expression in HL lymph node stroma and HL cells. Digital images of lymph node sections of the HL 23014-18 were acquired using the Aperio ScanScope Slide program of the AperioAT2 Scanner at 20× magnification. Panel (**A**): Subpanel a: Mallory trichrome images. Subpanel (**b**): IHC with the anti-TG2 mAb followed by GAR and DAB. Subpanel c: IHC with the anti-LOX rabbit ab174316 mAb. Panel (**B**): Layers of each virtual Mallory slide (**a**) or TG2 (**b**) or LOX (**c**) IHC stained slides deconvoluted with the automated image analysis algorithm (Aperio version 9.1). Insets in A and B: images of the whole section with the square indicating the region of enlargement. Panel (**C**): digital images of sections double stained with the anti-CD30 BerH2 mAb (subpanels (**a**,**b**)) or the anti-MUM-1 mAb (subpanel (**c**)) followed by the specific GAM and DAB (brown) and anti-LOX rabbit ab174316 mAb followed by GAR and Via Green Chromogen (green). Subpanels (**d**,**e**): enlargement of the white squares in subpanels (**Ca**,**Cb**).

**Figure 6 cancers-14-00259-f006:**
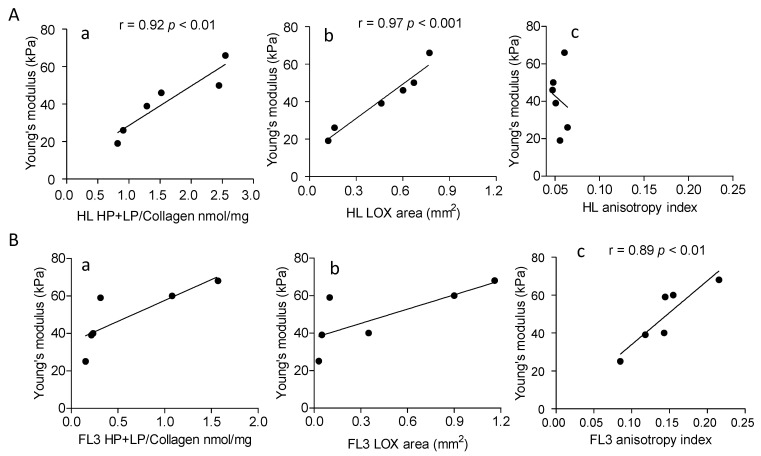
Correlation between Young’s modulus and collagen cross-linking, LOX expression or anisotropy in HL and NHL FL3. 6 HL (Panel (**A**)) and 6 NHL FL3 (Panel (**B**)) were analyzed. Subpanels a: HL and NHL FL3A collagen cross-linking (HP + LP/collagen nmol/mg) was evaluated as described in the Methods section and data analyzed for the correlation with Young’s modulus (kPa) measured as described. Subpanels b: in situ LOX expression in LN from HL and NHL FL3 was calculated as reported in the Methods, expressed as mm^2^ of LOX^+^ area (analyzed with the Aperio algorithm and quantified by the ImageScope software) and compared to the Young’s modulus (kPa). Subpanels c: anisotropy index evaluated as described in the Methods was analyzed for the correlation with the Young’s modulus. Linear regression (r) and significance (*p*) were calculated (Pearson correlation coefficient r) with the Graph Pad Prism software 5.

**Figure 7 cancers-14-00259-f007:**
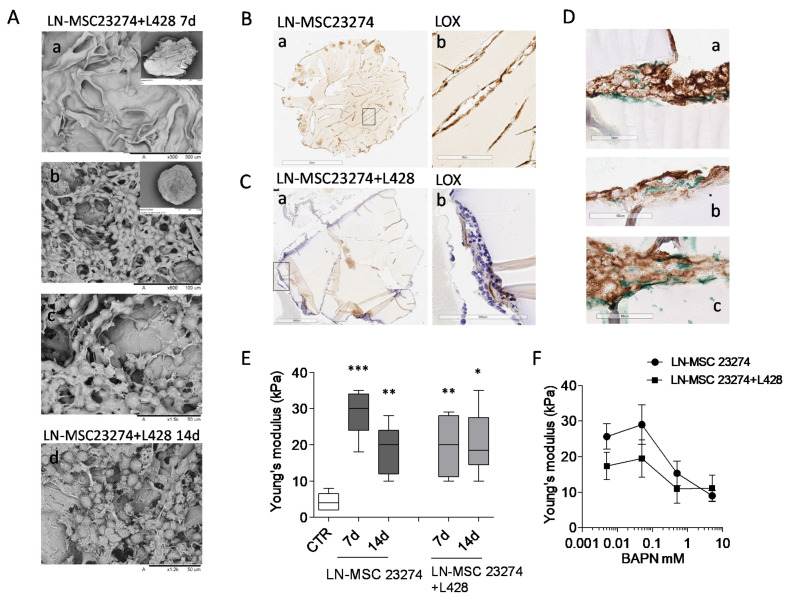
Involvement of LOX in GelMA stiffness. GelMA scaffolds repopulated with 2 × 10^5^ LN-MSC alone or with a mix of LN-MSC and 3 × 10^5^ L428 cells underwent SEM (Panel (**A**), mixture only) or IHC (Panel (**B**–**D**)) or elastometry to determine Young’s modulus at 7 or 14 d (Panel (**E**)). Parallel samples were exposed to serial dilution of the LOX inhibitor BAPN and Young’s modulus (kPa) was measured at 7 d (Panel (**F**)). Panel (**A**): SEM images of empty (**a**) or repopulated ((**b**) enlarged in (**c**) gelatin scaffolds at 7 d or 14 d (**d**); insets in a and b show the whole scaffold. Panel (**B**,**C**): sections of GelMA scaffolds repopulated with either LN-MSC (**B**) or a mixture of LN-MSC and L428 cells (**C**) stained with the anti-LOX rabbit ab174316 mAb followed by the specific GAM and DAB (in (**C**): counterstaining with HE). Subpanels b: enlargement of the rectangles in subpanels a. Panel (**D**): sections of GelMA scaffolds repopulated as in (**C**), double-stained with the anti-CD30 BerH2 mAb followed by the specific GAM and DAB (brown) and the anti-LOX rabbit ab174316 mAb followed by GAR and Vina Green Chromogen (green) in different scaffolds areas (subpanels (**a**–**c**)). Panel (**E**,**F**): Young’s modulus (kPa) of GelMA scaffolds either empty (CTR in (**E**)) or repopulated with LN-MSC alone or a mixture with L428 cells and cultured for 7 or 14 d (Panel (**E**)) or for 7 d with the LOX inhibitor BAPN at logarithmic dilutions from 5 mM to 50 µm or 0 mM (vehicle) concentrations (Panel (**F**)). * *p* < 0.05, ** *p* < 0.01, *** *p* < 0.001 vs. CTR. Mean ± SEM of three experiments performed in triplicate.

**Table 1 cancers-14-00259-t001:** Characteristics of lymphomas analyzed.

Histological Type ^1^	Sex	Age (Range)	Number of Cases
HL	6F 5M	28–69	11
FL1-2	4F 2M	36–78	6
FL3A	3F 7M	40–79	10
DLBCL	2F 6M	49–81	8
LDN	5F 5M	40–76	10

^1^ HL: Hodgkin lymphoma; FL1-2: non-Hodgkin follicular lymphoma grade 1–2; FL3A: non-Hodgkin follicular lymphoma grade 3A; DLBCL: diffused large B cell lymphoma; LDN: lymphadenitis. F: female; M: male. FL stages were according to the 2008 WHO classification [28].

## Data Availability

Not applicable.

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
