# Peer review of "Lysyl-Oxidase Dependent Extracellular Matrix Stiffness in Hodgkin Lymphomas: Mechanical and Topographical Evidence"

_cancers, 2022, doi:10.3390/cancers14010259_

Round 1
Reviewer 1 Report
In this article, the authors analyze how changes in ECM depend on the histotype of the lymphoma and report an increase in lymph node stiffness (LN) in human lymphomas, as measured by LN elastometry or by computerized imaging of biopsy specimens. Based on these data, digital pathology may contributes to making a correct diagnosis by distinguishing between HL and the other forms described and lysyl oxidase may represent a target for therapy in Hodgkin lymphomas.
- The drafting of the article is clear in all its components
- The materials and methods are well described in the different sections respecting the guidelines for the authors.
- The results were attenuated respecting the statistics and with an accurate description of the images obtained. Furthermore, the latter respect the requirements of the magazine in terms of format and style.
- The discussion is well articulated, clear and richly compared with the literature.
Author Response
We are pleased for the positive judgement of the reviewer.
Reviewer 2 Report
Percentile of overall quality: top 80th percentile of papers
SUMMARY
In this study the authors explored the mechanical properties of lymphomas as related to stage and histology. The idea of objectively studying tumor stiffness and correlating it to lymphoma clinical and biological properties is an interesting one with little reported in the literature with respect to hematopoetic malignancies, supporting the novelty of this work. They used digital pathology, SEM, collagen cross-linking and anisotropy evaluation, and young’s modulus assessment to help describe this phenomena. The authors use the LOX-inhibitor BAPN to inhibit the ability to increase stiffness within 3D tissue model. The work is novel and interesting, bridging engineering/material science principles with clinical features of cancer. There are a few major concerns regarding the overall findings, which are compelling in Hodgkins but don’t explain why DLBCL are more similar to benign LNs than to high grade LNs. A more molecular focus would increase the impact of the study.
MAJOR COMMENTS
The authors use beta-aminoproprionitrile (BAPN) as a LOX inhibitor to suggest that LOX is the specific enzyme responsible for extracellular matrix stiffness in Hodgkins lymphomas. However, how specific is this drug for LOX inhibition? BAPN may have other poorly understood toxic affects in human cells. Confirming LOX as the enzyme responsible for ECM stiffness could be confirmed far more compellingly using standard molecular biology techniques – knock down/out or overexpression for example. As described in the methods, BAPN is toxic to the cells at higher concentrations, and therefore the effect of BAPN may be just general cell toxicity/slower growth etc. There are no data showing that the BAPN does not cause other toxic effects on the cells at the concentrations used to demonstrate the impact on Young’s modulus.
The study reports that compared to benign lymph nodes, Hodgkins and advanced FL (3A) are more stiff. However, NHL often occur on a spectrum from low grade FL to intermediate grade FL, to high grade FL/transformed/DLBCL. Therefore, the observation that the DLBCL are less stiff than high grade FL is confusing and seems inconsistent with an overall model of increasing aggressiveness of NHL.
There is no discussion of immune infiltrates and how they might impact stromal properties. Hodgkins, for example, is thought to be composed of few cancer cells relative to the total tumor composition.
Was there any clear relationship with BCL2/6 and MYC status on tumor stiffness?
MINOR
“The first evidence provided by this work, is that different mechanical, topographical and architectural modifications of ECM are detectable in human lymphomas and can be matched to their histotype and grading.” – is this a typo? Did the authors mean to state “This is the first evidence….”.
Author Response
We are pleased for the overall favorable evaluation of our work and thank the reviewer for the constructive criticisms raised.
MAJOR COMMENTS
- The authors use beta-aminoproprionitrile (BAPN) as a LOX inhibitor to suggest that LOX is the specific enzyme responsible for extracellular matrix stiffness in Hodgkins lymphomas. However, how specific is this drug for LOX inhibition?
BAPN is sold as specific, non-selective LOX inhibitor by the producer (Sigma-Aldrich), as reported in the data sheet, although it is not shown the IC50 for each member of the family (LOX, LOXL1, LOXL2, LOXL3, LOXL4) and it can cross-react with other amine-oxidases. Nevertheless, BAPN is widely used to block LOX in vitro and in vivo [refs. 36-38 and 10.1038/ncomms14909 or 10.3390/antiox10020312]; this is reported is more than 100 papers and summarized in a recent review (https://pubmed.ncbi.nlm.nih.gov/33669630/). Its effects are dependent on the binding to the active site of LOX, resulting in the blocking of collagen cross-linking. In our model, IC50 of LOX inhibition evaluated with a fluorimetric assay was about 50µM (Suppl.Fig.3C and D).
- BAPN may have other poorly understood toxic affects in human cells….
We excluded the main toxic effects in our model: both viability and metabolism are preserved in MSC and HL cells upon exposure to BAPN from 50µM to 10mM up to 8 days of culture. See Suppl. Fig.3 and also the reply to point 4.
- Confirming LOX as the enzyme responsible for ECM stiffness could be con-firmed far more compellingly using standard molecular biology techniques – knock down/out or overexpression for example.
We agree with the reviewer that LOX knock down might be an elegant way to confirm once again that LOX is involved in matrix stiffness, due to increased collagen cross-linking. Nevertheless, this is already reported in the literature for many diseases [refs.36-38 and 40-43 as examples] and it was the starting point to check its expression in HL and inhibit its function in a 3D model.
- As described in the methods, BAPN is toxic to the cells at higher concentrations, and therefore the effect of BAPN may be just general cell toxicity/slower growth etc. There are no data showing that the BAPN does not cause other toxic effects on the cells at the concentrations used to demonstrate the impact on Young’s modulus.
The doses of BAPN showing toxicity are not cited in the Methods. In the Results section (lines 435-440) it is reported that:
“Titration experiments were performed, since BAPN effective concentrations reported in the literature varied from 10mM to 10µM, depending on cell type (solid tumors or cancer associated fibroblasts) and on culture conditions (conventional cultures vs spheroids) [36-38]. The concentrations chosen for our experimental setting (from 5 to 0.1mM) were not toxic for either LN-MSC or L428 cells (Suppl. Fig.S3A and B), but were efficient in inhibiting LOX activity in both cell types (Suppl. Fig.S3, C and D).”
It has to be noted that viability was evaluated also as ATP content and glucose consumption has been found superimposable to that of untreated cells, either MSC or L428, up to 8 days of culture (the time chosen to evaluate the effects of BAPN on Young’s modulus was day 7), supporting that cells are not only viable but also metabolically active.
- …NHL often occur on a spectrum from low grade FL to intermediate grade FL, to high grade FL/transformed/DLBCL. Therefore, the observation that the DLBCL are less stiff than high grade FL is confusing and seems inconsistent with an overall model of increasing aggressiveness of NHL.
That DLBCL represents a distinct entity among NHL or a more aggressive stage of FL is still under debate, since FL and DLBCL coexist in some patients at the onset of disease, in other patients DLBCL feature arise after a period of time. In most cases, DLBCL is treated as FL3B (that are not considered in our study). In our study, we considered only patients with distinct histological features at diagnosis. DLBCL Young’s modulus was significantly higher than that of LDN and FL1-2, intermediate between FL3A and HL (Fig.3A). No differences in collagen cross-linking (increased in HL) or anisotropy (altered in NHL FL3) was found, but a significant alteration of the P/A ratio in electron microscopy images is reported in Fig.2C (see below for discussion of this point). We tried to address this point in the Discussion (lines 473-477):
“possible explanations for this observation: first, stiffness in DLBCL is not directly related to LN stroma, but rather to the degree of lymphoma cell infiltration (overcrowding and indirect effect on the stroma due to cell traction) within the LN, leading to disruption of the follicular structures in some areas as occurs in solid tumors [41]. Other collagen abnormalities, different from cross-linking and not detectable with polarized light, are responsible for shape alterations in DLBCL or other matrix proteins are damaged/modified. In any case, elastometry can evidence these alterations in all aggressive lymphomas.”
- There is no discussion of immune infiltrates and how they might impact stromal properties. Hodgkins, for example, is thought to be composed of few cancer cells relative to the total tumor composition.
We agree with the reviewer that this topic is very relevant in the study of all neoplasias. A paragraph has been included in the Discussion (lines 497-504) and two ref. added [42,43]:
“Different cell types, including inflammatory cells, can condition tumor environment and influence ECM modifications [42]; this is particularly true in HL, where neoplastic cells often represent a minority of the tumor population [21, 23-25]. How much is the relative contribution of the immune infiltrate, of the resident non-transformed cells and that of tumor cells is difficult to be assessed. Metalloproteases or inflammatory cytokines, such as tumor necrosis factor-α or transforming growth factor-β, frequently found in HL microenvironment, are reported regulators of ECM remodeling, directly or indirectly influencing LOX activity and collagen cross-linking [21, 25, 43].”
- Was there any clear relationship with BCL2/6 and MYC status on tumor stiffness?
This is a very compelling and challenging question: the relationship between multiple chromosomal translocation/aberrations in double/triple hit lymphomas and their evolution towards aggressive clinical forms as high-grade lymphomas is well-known. In our study however, no MYC/Bcl2/Bcl6 translocation were found in DLBCL, so that this interesting issue remains open.
MINOR POINTS
“The first evidence provided by this work, is that different mechanical, topographical and architectural modifications of ECM are detectable in human lymphomas and can be matched to their histotype and grading.” is this a typo? Did the authors mean to state “This is the first evidence….”.
The sentence has been rephrased as follows: “This work provides evidence that….
Reviewer 3 Report
The manuscript by Alfano et al describes mechanical and topographical evidence of Lysyl-oxidaze dependent extracellular matrix stiffness in Hodkin lymphomas. The experiments are straight-forward and demonstrates elastic modulus is higher in HL than NHL LN, also reflecting the lymphoma stage grade, as a consequence of collagen cross-linking together with architectural modifications related to increased anisotropy and Lysyl-oxidaze increase. The work is an extension of what is described for solid tumors. Experimental approaches are similar to that of described for solid tumors. The report provides information that different mechanical, topographical and/or architectural modifications of ECM are detectable in human lymphomas and are related to their histotype and grading. However, there is always a concern that when tissues taken out, that is removed from the appropriate in vivo environment would there be alteration in mechanical, topographical and/or architectural that would skew the results. Nevertheless, the assays can be supporting evidence along with other parameters for diagnosis, staging and therapeutic applications.
Author Response
We thank the Reviewer for the positive judgement of our work.
As for the criticism: "...However, there is always a concern that when tissues taken out, that is removed from the appropriate in vivo environment would there be alteration in mechanical, topographical and/or architectural that would skew the results"....we added a sentence to the discussion on this limitation to be taken into account:
"Although it should be taken into account that tissues removed from the in vivo environment would undergo alterations in their physical and mechanical properties, the reported differences are referred to samples of lymphoma specimens always compared to non neoplastic lymph nodes in the same conditions."
We also cheched the manuscript for spelling errors throughout the text.
Round 2
Reviewer 2 Report
the responses are adequate and I feel the paper acceptable for publication.